# REAL-WORLD BENCHMARKS MAKE MEMBERSHIP INFERENCE ATTACKS FAIL ON DIFFUSION MODELS

## ABSTRACT

Membership inference attacks (MIAs) on diffusion models have emerged as potential evidence of unauthorized data usage in training pre-trained diffusion models. These attacks aim to detect the presence of specific images in training datasets of diffusion models. Our study delves into the evaluation of state-of-the-art MIAs on diffusion models and reveals critical flaws and overly optimistic performance estimates in existing MIA evaluation. We introduce CopyMark, a more realistic MIA benchmark that distinguishes itself through the support for pre-trained diffusion models, unbiased datasets, and fair evaluation pipelines. Through extensive experiments, we demonstrate that the effectiveness of current MIA methods significantly degrades under these more practical conditions. Based on our results, we alert that MIA, in its current state, is not a reliable approach for identifying unauthorized data usage in pre-trained diffusion models. To the best of our knowledge, we are the first to discover the performance overestimation of MIAs on diffusion models and present a unified benchmark for more realistic evaluation.

## 1 INTRODUCTION AND RELATED WORKS

Diffusion models (Sohl-Dickstein et al., 2015; Song & Ermon, 2019; Ho et al., 2020; Song et al., 2020b) have revolutionized the field of image synthesis. A notable advantage of these models is their ability to train stably on vast web-sourced datasets containing billions of images (Schuhmann et al., 2021; 2022). This capability has paved the way for large-scale pre-trained models in image synthesis (Rombach et al., 2022; Podell et al., 2023; Chen et al., 2023; Esser et al., 2024). However, these pre-trained models have raised concerns regarding unauthorized data usage (Samuelson, 2023; Sag, 2023), as their training datasets often include numerous copyrighted images without proper authorization. In response, copyright owners have initiated a series of lawsuits against producers of pre-trained diffusion models (Andersen, 2023; Zhang, 2024). Within this context, *membership inference attacks* (**MIA**s) on diffusion models (Duan et al., 2023; Kong et al., 2023; Fu et al., 2023; 2024; Tang et al., 2023) have emerged. MIAs aim to separate *members* (data used for training) and *non-members* (data not used for training). Their results help determine whether specific images were included in the training dataset of diffusion models, thus being considered as potential evidence of unauthorized data usage in AI copyright lawsuits related to diffusion models.

However, recent research suggests that MIAs on Large Language Models (LLMs) may perform successfully because they are evaluated under defective setups with distribution shift (Das et al., 2024; Maini et al., 2024). Their performance is confounded by evaluating on non-members belonging to a different distribution from the members (Maini et al., 2024). This finding raises concern on the true effectiveness of MIAs, including those on diffusion models.

Inspired by the above idea, we investigate the current evaluation of MIAs on diffusion models. Unfortunately, we find that there are similar defects in the evaluation of MIAs on diffusion models. Specifically, the evaluation is based on 1) over-trained models (Duan et al., 2023; Kong et al., 2023; Fu et al., 2023; Pang et al., 2023) and 2) member datasets and non-member datasets with distribution shifts (Duan et al., 2023; Kong et al., 2023). Both setups make the task of MIA easier than the real-world scenario, with pre-trained diffusion models and unshifted members and non-members. This defective evaluation leaves unknown the true performance of MIAs on diffusion models.

To fill the blank of real-world evaluation, we build CopyMark, the first unified benchmark for membership inference attacks on diffusion models. CopyMark gathers 1) all pre-trained diffusion

models (Rombach et al., 2022; Gokaslan et al., 2023; YEH et al., 2023) 2) with accessible unshifted non-member datasets (Dubiński et al., 2024; Gokaslan et al., 2023; YEH et al., 2023). We implement state-of-the-art MIA methods on these diffusion models and datasets. To refine the current evaluation pipeline, we introduce extra *test datasets* in addition to the original *validation datasets* (datasets where we find the optimal threshold to separate members and non-members) and use these test datasets for blind test of MIAs. Through extensive experiments, we show that MIA methods on diffusion models suffer significantly bad performances on our realistic benchmarks. Our result alerts the fact that current MIAs on diffusion models only appear successful on unrealistic evaluation setups and cannot perform well in real-world scenarios. Our contributions can be summarized as follows:

- We reveal two fatal defects in current evaluation of MIAs on diffusion models: over-training and dataset shifts (Section 3).

- We design and implement CopyMark, a novel benchmark to evaluate MIAs on diffusion models in real-world scenarios. To the best of our knowledge, this is the first unified benchmark for MIAs on diffusion models (Section 4).

- We are the first to alert that the performance of MIAs on diffusion models has been overestimated through extensive experiments (Section 5). This is significant both theoretically for future research in this area and empirically for people involving in the AI copyright lawsuits who expect MIAs as evidence .

## 2 BACKGROUND

### 2.1 DIFFUSION MODELS

Diffusion models (Sohl-Dickstein et al., 2015; Song & Ermon, 2019; Ho et al., 2020; Song et al., 2020b; Dhariwal & Nichol, 2021) achieve state-of-the-art performance in generative modeling for image synthesis. Diffusion models are latent variable models in the form of $p_\theta(x_{0:T})$ with latent variables $x_{1:T}$ sharing the same shape with data $x_0 \sim q(x_0)$ (Ho et al., 2020; Song et al., 2020a). $p_\theta(x_{0:T})$ is denoted by *reverse process* since it samples $x_{t-1}$ by progressively reversing timestep $t$ with $p_\theta(x_{t-1}|x_t)$.

$$p_\theta(\boldsymbol{x}_{0:T}) = p(\boldsymbol{x}_T) \prod_{t \geq 1} p_\theta(\boldsymbol{x}_{t-1}|\boldsymbol{x}_t), p_\theta(\boldsymbol{x}_{t-1}|\boldsymbol{x}_t) := \mathcal{N}(\boldsymbol{x}_{t-1}; \mu_\theta(\boldsymbol{x}_t, t), \Sigma_\theta(\boldsymbol{x}_t, t)) \quad (1)$$

with $p(x_T)$ as the prior and set as standard Gaussian. Diffusion models are distinguished by its posterior $q(x_{1:T}|x_0)$ which is a Markov process that progressively adds gaussian noise to the data, termed by the *forward process* (Ho et al., 2020).

$$q(\boldsymbol{x}_{1:T}|\boldsymbol{x}_0) = \prod_{t \geq 1} q(\boldsymbol{x}_t|\boldsymbol{x}_{t-1}), q(\boldsymbol{x}_t|\boldsymbol{x}_{t-1}) := \mathcal{N}(\boldsymbol{x}_t; \sqrt{1-\beta_t}\boldsymbol{x}_{t-1}, \beta_t \boldsymbol{I}) \quad (2)$$

The model is trained by matching the reverse step $p_\theta(\boldsymbol{x}_{t-1}|\boldsymbol{x}_t)$ with the forward step $q(\boldsymbol{x}_{t-1}|\boldsymbol{x}_t, \boldsymbol{x}_0)$ conditioned on data $\boldsymbol{x}_0$, where the model learns to sample $x_0$ from the prior $p(x_T)$ by progressively sampling $x_{t-1}$ from $x_t$ with $p_\theta(x_{t-1}|x_t)$.

Diffusion models are scalable for pre-training over large-scale text-image data (Rombach et al., 2022; Podell et al., 2023; Chen et al., 2023; Luo et al., 2023; Esser et al., 2024). However, pre-trained diffusion models may include copyright images in the training dataset without authorization (Andersen, 2023; Zhang, 2024). This unauthorized data usage now raises critical ethics issues.

In this paper, we focus on one family of diffusion models, Latent Diffusion Models (LDMs) (Rombach et al., 2022). LDMs are the base architecture of state-of-the-art pre-trained diffusion models (Rombach et al., 2022; Podell et al., 2023; Chen et al., 2023; Luo et al., 2023; Esser et al., 2024). These models are the origin of the above unauthorized data usage. To investigate whether MIAs on diffusion models are effective tools in detecting unauthorized data usage, we evaluate them on these LDM-based pre-trained diffusion models.

## 2.2 Membership Inference Attacks on Diffusion Models

Membership Inferences Attacks (MIAs) (Shokri et al., 2017; Hayes et al., 2017; Chen et al., 2020; Carlini et al., 2023) determine whether a datapoint is part of the training dataset of certain diffusion models. We give the formal problem statement of MIAs as follows:

**Membership Inference Attacks** *Given the training dataset $\mathcal{D}_{member}$ (members) of a model $\theta$ and a hold-out dataset $\mathcal{D}_{non}$ (non-member), membership inference attacks aim at designing a function $f(x, \theta)$, that $f(x, \theta) = 1$ for $x \in \mathcal{D}_{member}$ and $f(x, \theta) = 0$ for $x \in \mathcal{D}_{non}$. Here, $x$ is an image.*

Currently, they are two types of MIA methods on diffusion models:

- Loss-based MIAs (Duan et al., 2023; Fu et al., 2023; Kong et al., 2023): Loss-based MIAs are built on the general hypothesis that the training loss of members is smaller than that of non-members. They therefore calculate a function $R(x, \theta) \in \mathbb{R}$ for data point $x$ based on the training loss of diffusion models and find an optimal threshold $\tau$ to discriminate $R(x, \theta)$ of members and non-members.

$$f(x, \theta) := \mathbb{1}[R(x, \theta) < \tau] \quad (3)$$

- Classifier-based MIAs (Pang et al., 2023): Classifier-based MIAs believe that members and non-members have different features $g_\theta(x)$ in diffusion models. The features could be gradients and neural representations. These features can be used to train a neural network $F$ to classify members and non-members.

$$f(x, \theta) := F(g_\theta(x)) \quad (4)$$

Membership inference attacks of diffusion models are evaluated by two metrics: true positive rate at a low false positive rate (Carlini et al., 2023) and AUC. Here, true positive rate (TPR) means the percentage of predicting members as members correctly, while false positive rate (FPR) means the percentage of predicting non-members as members falsely. We are interested in this *TPR at $X\%$ FPR* because we only care about whether some members could be identified without errors in practice. Take detecting unauthorized data usage as an example. With only one copyright image is determined as the member, we can then prove the existence of unauthorized data usage. We also include AUC as our metric, which is a classical measurement on the discrimination of MIA methods. In practice, we calculate TPRs and FPRs on the dataset. Then, we search for the optimal threshold and calculate AUC on the same dataset.

Notably, the evaluation of MIAs need to follow the *MI security game protocol* (Carlini et al., 2023; Hu & Pang, 2023), that the member $\mathcal{D}_{member}$ and the non-member $\mathcal{D}_{non}$ should come from the same data distribution. This is because we do not know the distribution of members and non-members in real-world scenarios of MIAs, for example, detecting unauthorized data usage in pre-trained diffusion models. We then need to assume the worst case when the member and the non-member $\mathcal{D}_{non}$ come from the same data distribution so that we cannot simply distinguish them without the help of model $\theta$. However, we will show in the rest of this paper that this protocol is not well followed in existing MIAs on diffusion models.

## 3 Evaluation of MIAs on Diffusion Models Are Defective

In this section, we reveal the fundamental defect within current evaluation of MIAs on diffusion models. We start from explaining two choices in the evaluation setup of MIAs:

- over-training v.s. pre-training: Over-training refers to overly training a model for large epochs on a small training dataset, e.g. 2048 epochs on CIFAR10. What distinguishes over-training from traditional over-fitting is that over-training improves the generation performance of small diffusion models and is considered as a default setup for this training (Song & Ermon, 2020). That explains why most existing diffusion MIA benchmarks have the problem of over-training. Pre-training, in contrast, only trains diffusion models on a very large dataset, for example, for 1 or 2 epochs.
- shifted datasets v.s. unshifted datasets: Dataset shift means that the member dataset and the non-member dataset do no come from the same data distribution. On the contrary, unshifted

Table 1: List of all existing benchmark setups of diffusion MIA. All these setups suffer from over-training or dataset shifts or both. Notably, the choice of models and datasets are not the necessary and sufficient conditions of having over-training or dataset shifts, while they are listed only as references.

| Model | DDPM | DDPM | DDPM | DDPM | DDPM | LDM | LDM | LDM | SD1.5 |
| Dataset | CIFAR-10 | CIFAR-100 | ImageNet | CelebA | COCO | COCO | Pokemon | CelebA | LAION |
|---|---|---|---|---|---|---|---|---|---|
| SecMI | ✓ | ✓ | ✓ | ✗ | ✗ | ✓ | ✓ | ✗ | ✓ |
| PIA | ✓ | ✓ | ✓ | ✗ | ✗ | ✗ | ✗ | ✗ | ✓ |
| PFAMI | ✗ | ✗ | ✓ | ✓ | ✗ | ✗ | ✓ | ✓ | ✗ |
| GSA | ✓ | ✗ | ✓ | ✗ | ✓ | ✗ | ✗ | ✗ | ✗ |
| Over-training | ✓ | ✓ | ✓ | ✓ | ✓ | ✓ | ✓ | ✓ | ✗ |
| Shifted Datasets | ✗ | ✗ | ✗ | ✗ | ✗ | ✗ | ✗ | ✓ | ✓ |

| Model | Member | Non-member | Dataset Shift | Dataset Size (k) | Epochs | Over-training |
|---|---|---|---|---|---|---|
| DDPM | CIFAR-10 | CIFAR-10 | ✗ | 25/8 | 4096/400 | ✓ |
| DDPM | ImageNet | ImageNet | ✗ | 50/30/8 | 300/500/400 | ✓ |
| LDM | CelebA | FFHQ | ✓ | 50 | 500 | ✓ |
| SD1.5 | LAION | MS-COCO | ✓ | ~600,000 | 1 | ✗ |

member and non-member datasets refer to the condition that the member dataset and the non-member dataset come from the same data distribution.

Over-training and shifted datasets are the unrealistic choice. Specifically,

- over-training easily gives rise to over-fitting, since the model is trained for hundreds of steps on each data point from a limited dataset. This over-fitting markedly lowers the training loss of members and even causes memorization (Gu et al., 2023), making it easier to distinguish members from non-members based on training losses. However, recent progress in pre-training diffusion models shows the potential to train photorealistic diffusion models for only 1 epoch on large-scale text-image datasets (Rombach et al., 2022; Gokaslan et al., 2023), which does not make the training loss of members much lower than that of non-members (Wen et al., 2024). Since these pre-trained models are the real-world interests of MIAs on diffusion models, evaluting MIAs on over-trained diffusion models are unrealistic.

- dataset shift makes it possible to distinguish members from non-members without accessing the model. Hence, MIA methods succeeding on shifted datasets are probably dataset classifiers (Liu & He, 2024) that only captures the difference in image semantics, rather than real membership inference attacks. Such dataset classifiers will fail on correctly discriminating members and non-members that come from the same data distribution.

Table 1 examines the evaluation setup of MIAs on diffusion models from the perspective of over-training and shifted datasets and details some commonly used setups. Unfortunately, we find that there are either over-training or shifted datasets or both in these setups. For example, DDPM + CIFAR-10 (member) & CIFAR-10 (non-member) (Duan et al., 2023; Fu et al., 2024; Pang et al., 2023), DDPM + ImageNet (member) & ImageNet (non-member) (Duan et al., 2023; Fu et al., 2023; Kong et al., 2023; Pang et al., 2023), and LDM + CelebA (member) & FFHQ (non-member) (Fu et al., 2023) over-train diffusion models for at least 300 iterations on each of the data point. On the other hand, although SD1.5 + LAION (member) & MS-COCO (non-member) (Duan et al., 2023; Kong et al., 2023) exploits a pre-trained diffusion model (Rombach et al., 2022), it picks non-members from MSCOCO, whose distribution is markedly different from that of LAION. With over-training and dataset shift, it is unknown whether the success of MIA methods depends on the over-fitting or a non-member dataset whose distribution differs from that of the member dataset. This is a fatal defect of current evaluation of MIAs on diffusion models.

While avoiding over-training is simple by using pre-trained diffusion models for benchmarks, quantifying and preventing dataset shifts are not trivial. We will then conduct experiments to measure dataset shifts and show whether our benchmark eliminates dataset shifts (Section 5.1). We investigate the impact of different levels of dataset shifts on MIA performance (Section 5.3).

Table 2: Five evaluation setups in CopyMark. **(a)** and **(b)** are defective setups from previous evaluation. **(c)**, **(d)**, and **(e)** are novel setups with no over-training and minor or no dataset shift.

| Setup | Model | Member | Non-member | Dataset Shift | Dataset Size (k) | Epochs | Over-training |
|-------|-------|--------|------------|---------------|------------------|--------|---------------|
| (a) | LDM | CelebA | FFHQ | ✓ | 50 | 500 | ✓ |
| (b) | SD1.5 | LAION | MS-COCO | ✓ | ~600,000 | 1 | ✗ |
| (c) | SD1.5 | LAION | LAION | ✗ | ~600,000 | 1 | ✗ |
| (d) | CommonCanvas-XL | CommonCatalog | MS-COCO | ✓(minor) | ~2,500 | 1 | ✗ |
| (e) | Kohaku-XL | Hakubooru | Hakubooru | ✗ | ~5,200 | 1 | ✗ |

## 4 COPYMARK: REAL-WORLD BENCHMARK FOR DIFFUSION MIAS

To overcome the defect in previous evaluation and investigate the real-world performance of MIAs on diffusion models, we design and implement CopyMark, the first unified benchmark for MIAs on diffusion models. CopyMark distinguishes itself from previous evaluation from the following three aspects: **1)** CopyMark is built on pre-trained diffusion models with no over-training and member and non-member datasets without dataset shift, which overcome these two defects of previous evaluation (Section 4.1); **2)** CopyMark conducts blind evaluation on a test dataset other than the validation dataset used to find the threshold or train the classifier, which examines how MIAs perform on a blind test (Section 4.2); **3)** CopyMark is implemented on *diffusers*, the state-of-the-art inference framework for diffusion models, making it flexible to generalize to new diffusion models (Section 4.3)

### 4.1 MODELS AND DATASETS

Pre-trained diffusion models are the real-world interests of MIAs on diffusion models. Hence, we construct CopyMark on these pre-trained diffusion models. This covers the defect of over-training. However, it is non-trivial to select proper models for the evaluation of MIAs, because they must meet the following two requirements: 1) The training dataset (member dataset) is accessible to the public, and 2) There exist candidate non-member datasets whose distributions are similar or identical to that of the training dataset. We find three pre-trained diffusion models that meet the above requirements. We detail these models and the choice of their member and non-member datasets as follows:

- **Stable Diffusion v1.5** (Rombach et al., 2022): The most widely-used pre-trained diffusion model. Stable Diffusion v1.5 is trained for 1 epoch on LAION Aesthetic v2 5+ (Schuhmann et al., 2021; 2022). We follow Dubiński et al. (2024) to choose LAION Multi Translated as the source of non-members. There is no dataset shift since both member and non-member datasets come from the same distribution of LAION-2B dataset. We denote this setup by **(c)**.

- **CommonCanvas-XL-C** [1] (Gokaslan et al., 2023): A pre-trained diffusion model in the architecture of SDXL (Podell et al., 2023). CommonCanvas-XL-C is trained for 1 epoch on CommonCatalog (Gokaslan et al., 2023), a large dataset consisting of multi-source Creative Commons licensed images. CommonCanvas-XL-C uses MS-COCO2017 as its validation dataset for generation performance. This inspires us to pick its non-members from MS-COCO2017. However, these member and non-member datasets have dataset shift because of the distribution difference between CommonCatalog and MS-COCO2017. We will show that this shift is minor in Section 5.1. We denote this setup by **(d)**.

- **Kohaku-XL-Epsilon** [2] (YEH et al., 2023): An SDXL fine-tuned on 5.2 millions of comic images from HakuBooru dataset (YEH et al., 2023) for 1 epoch. We follow the instruction in the homepage to separate HakuBooru dataset into the training dataset and the rest hold-out dataset. Then, we pick members from the training dataset and non-members from the hold-out dataset. As members and non-members are randomly picked from the same dataset, there is no dataset shift in Kohaku-XL-Epsilon. We denote this setup by **(e)**.

Additionally, we also implement two previous defective setups in CopyMark: **(a)** LDM + CelebA (member) & FFHQ (non-member) (Fu et al., 2023) and **(b)** LDM + LAION (member) & MS-COCO2017 (non-member) (Duan et al., 2023; Kong et al., 2023). These two setups serve as a

---

[1] https://huggingface.co/common-canvas/CommonCanvas-XL-C
[2] https://huggingface.co/KBlueLeaf/Kohaku-XL-Epsilon

reference to our three new setups and also validate the correct of our implementation of MIA methods. The sanity check of datasets is discussed in Appendix A.1. All setups are summarized in Table 2.

**Are there over-training and dataset shifts?** Our choices of datasets and models in setup **(c)**, **(d)**, and **(e)** are dedicated to eliminate over-training and dataset shifts. Specifically, all three pre-trained diffusion models are trained for only 1 epoch on the dataset, which is the minimum number of training epochs. Hence, there is no over-training. As for dataset shifts, we conduct experiments in Section 5.1 to validate that there are at most only minor dataset shifts between members and non-members.

## 4.2 Two-stage Evaluation with Validation Datasets and Test Datasets

MIA methods on diffusion models use one dataset to find the optimal threshold or train the classifier (see Section 2.2 for explanation). However, previous evaluation of MIA methods also uses the same dataset for evaluation. This raises doubts on the generalizability of these MIA methods. We propose to complement this drawback by introducing an extra dataset called test dataset.

Specifically, CopyMark randomly picks two groups of data with the same number of members and non-members from the source. We denote one by the validation dataset and the other by the test dataset. Our evaluation pipeline has two stages. The first stage is the same as previous evaluation, that we use the validation dataset to search for the optimal threshold to calculate TPR at $X\%$ FPR and AUC or train the classifier. The second stage is different, that we test the optimal threshold or the trained classifier on the test dataset. Since the test dataset does not involve in searching the optimal threshold or training the classifier, the second stage can be viewed as a blind test to the threshold or the classifier. It is noticeable that we can only post the TPR and FPR by the optimal threshold or the trained classifier on the test dataset. We summarize two stages in Algorithm 1 and Algorithm 2.

---

**Algorithm 1** The first stage (previous)

**Input:** Evaluation dataset $\mathcal{D} = \{(x,y)\}$, FPR upper bound $X\%$
// $y = 1$: member, $y = 0$: non-member
**Output:** TPR, threshold $\tau^\star$
// score calculation
$Q = \emptyset$
**for** $(x,y) \sim \mathcal{D}$ **do**
  $r \leftarrow R(x,\theta)$
  $Q \leftarrow Q \cap \{(r,y)\}$
**end for**
// threshold optimization and evaluation
$\tau_{min} = \min_{(r,y)\sim Q} r$
$\tau_{max} = \max_{(r,y)\sim Q} r$
$\tau^\star := \arg\max_{\tau \in [\tau_{min},\tau_{max}]} \frac{|\{(r,y)\in Q | r \leq \tau \wedge y=1\}|}{|\{(r,y)\in Q | y=1\}|}$

$\text{TPR} := \max_{\tau \in [\tau_{min},\tau_{max}]} \frac{|\{(r,y)\in Q | r \leq \tau \wedge y=1\}|}{|\{(r,y)\in Q | y=1\}|}$

$s.t. \frac{|\{(r,y)\in Q | r \leq \tau \wedge y=0\}|}{|\{(r,y)\in Q | y=0\}|} \leq X\%$
**return** TPR, $\tau^\star$

---

**Algorithm 2** The second stage (new)

**Input:** Test dataset $\mathcal{D}' = \{(x,y)\}$, optimal threshold $\tau^\star$
// $y = 1$: member, $y = 0$: non-member
**Output:** TPR, FPR
// score calculation
$Q = \emptyset$
**for** $(x,y) \sim \mathcal{D}'$ **do**
  $r \leftarrow R(x,\theta)$
  $Q \leftarrow Q \cap \{(r,y)\}$
**end for**
// threshold evaluation

$\text{TPR} := \frac{|\{(r,y)\in Q | r \leq \tau^\star \wedge y=1\}|}{|\{(r,y)\in Q | y=1\}|}$

$\text{FPR} := \frac{|\{(r,y)\in Q | r \leq \tau^\star \wedge y=0\}|}{|\{(r,y)\in Q | y=0\}|}$

**return** TPR, FPR

---

## 4.3 Implementation

Previously, there is no unified benchmark for MIAs on diffusion models. To fill this blank, we implement all state-of-the-art baseline methods of MIAs on diffusion models in CopyMark. We base our implementation on diffusers (von Platen et al., 2022), the state-of-the-art inference framework for diffusion models. We discuss the details of our implemention by points:

**diffusers** diffusers provides a unified API for running different diffusion models. We implement MIA methods as *pipeline* objects in diffusers. This enables MIA methods to generalize swiftly to different diffusion models. While diffusers is updated to support newly released diffusion models, CopyMark can benefit from the update and provide straight-forward generalization of MIAs to new

diffusion models. This is the main advantage of implementing CopyMark on diffusers. We omit other implementation details to Appendix A.1.

**Evaluation Metrics** Following Duan et al. (2023), we randomlyy pick 2500 images as members and 2500 images as non-members. We repeat this picking twice to produce one validation dataset and one test dataset. For the validation dataset, we follow Carlini et al. (2023) post TPR at $1\%$ FPR and $0.1\%$ together with the AUC (Algorithm 1). For the test dataset, we post the TPR and FPR for two optimal thresholds (the threshold at $1\%$ FPR and that at $0.1\%$ FPR) obtained from the validation set (Algorithm 2). The random seed of all evaluation is fixed for full reproducibility.

**Baselines** Duan et al. (2023); Fu et al. (2023) show that general MIA methods do not work well in diffusion models. Hence, our baselines consist of MIA methods on diffusion models, including *SecMI* (Duan et al., 2023), *PIA* (Kong et al., 2023), *PFAMI* (Fu et al., 2023), *GSA$_1$* (Pang et al., 2023), and *GSA$_2$* (Pang et al., 2023). SecMI, PIA, and PFAMI are loss-based MIA methods, while GSA$_1$ and GSA$_2$ are classifier-based MIA methods. In addition to these MIA methods, we follow (Das et al., 2024) to implement a *Blind* baseline. This blind baseline trains a ConvNext (Liu et al., 2022) to classify members and non-members. We omit the details of baseline implementation to Appendix A.1.

## 5 EXPERIMENTS: DIFFUSION MIA FAILS ON REAL-WORLD SETUPS

In this section, we use experiments to demonstrate the failure of current diffusion MIA methods on real-world benchmarks and validate our attribution in Section 3. We first quantify dataset shifts of all benchmark setups in CopyMark and validate that there are no or only minor dataset shifts (Section 5.1). Then, we post the performance of MIA methods on differrent setups of CopyMark (Section 5.2), which reproduces their success on defective benchmarks and reveals their failure on our real-world benchmarks. To further investigate the impact of dataset shifts on the MIA performance, we apply current MIA methods on a series of non-member datasets with different proportions of shifted and non-shifted data, and find that dataset shifts and current MIA performance have strong correlation (Section 5.3). This validates our claim about the hallucination of success of diffusion MIA.

### 5.1 QUANTIFYING DATASET SHIFTS

We first validate the existence of dataset shifts between member datasets and non-member datasets by quantifying them. We use CLIP (Radford et al., 2021) to extract the representation of images in members & non-members and take them as the base of our quantification. Figure 2 demonstrates the visualization of semantic representations. Then, we exploit two strategies to quantifying dataset shifts: **1)** Explicitly, we calculate three distance metrics between the representation distribution of members and that of non-members: normalized Wasserstein distance (by the internal variance), Frechet distance, and Mahalanobis distance. **2)** Implicitly, we train a linear classifier to classify the representations of members & non-members.

Table 3 shows the result of **1)**. Distances between members and non-members are distinctly larger in two defective setups **(a)** and **(b)**, compared to our three improved setups **(c)**, **(d)**, and **(e)**. Similar observations appear in the result of **2)**, posted in Table 4. Linear classifiers can easily separate representations of members and those of non-members in two defective setups, which cannot be done in our three improved setups. Notably, both distance metrics and the classifier shows that there are more dataset shifts in our setup **(d)** than setup **(c)** and **(e)**, although these shifts are minor. Specifically, three distances are consistently larger for setup **(d)**, and the true positive rate of the linear classifier is little higher. This validates the description in Section 3 that there are minor dataset shifts in setup **(d)**.

Comparing Table 3 and Table 4, we have two conclusions: First, among three distance metrics, Frechet distance (FD) can better measure the dataset shifts, for it shows the best consistency with the performance of the linear classifier. This is also supported by the fact that Frechet Inception Distance (FID), a Frechet distance using Inception representations, is the most widely-used metrics for the shift between real data and data generated by generative models. Second, a golden FD for an excellent diffusion MIA benchmark should be around than $0.05$, while only our setup $(e)$ meets the requirement. In Section 5.2, we will see that setup $(e)$ disables all current diffusion MIA.

Table 3: Normalized Wasserstein distances (NWD), Frechet distances (FD), and Mahalanobis distances (MD) between CLIP representations of members and those of non-members of five setups in CopyMark. We use Red text to denote the defective setups **(a)** and **(b)**, which have higher distances conspicuously. In contrary, our three realistic setups **(c)**, **(d)**, and **(e)**, have much smaller distances, indicating they do not have distinguishable dataset shifts.

| Metrics | (a) | | (b) | | (c) | | (d) | | (e) | |
|---|---|---|---|---|---|---|---|---|---|---|
| | Val | Test | Val | Test | Val | Test | Val | Test | Val | Test |
| NWD | 5910 | 6376 | 3779 | 3647 | 1974 | 2022 | 3628 | 3528 | 3305 | 3598 |
| FD | 0.324 | 0.333 | 0.249 | 0.248 | 0.059 | 0.059 | 0.126 | 0.121 | 0.037 | 0.036 |
| MD | 15.47 | 16.88 | 7.07 | 7.11 | 1.39 | 1.50 | 5.04 | 5.03 | 1.26 | 1.23 |

Table 4: Using a linear classifier to classify CLIP representations of members and those of non-members. High TPRs and TNRs show that members and non-members in defective Setup **(a)** and Setup **(b)** can be easily separated, indicating there are severe dataset shifts in these two setups. In our novel setups **(c)**, **(d)**, and **(e)**, there are only minor or no dataset shifts, indicated by the TPRs and TNRs around $0.5$. Our new setups fit the real-world MIA scenario better.

| Setup | (a) | | (b) | | (c) | | (d) | | (e) | |
|---|---|---|---|---|---|---|---|---|---|---|
| | Val | Test | Val | Test | Val | Test | Val | Test | Val | Test |
| TPR | 0.953 | 0.946 | 0.880 | 0.905 | 0.540 | 0.534 | 0.690 | 0.606 | 0.586 | 0.543 |
| FPR | 0.037 | 0.073 | 0.182 | 0.154 | 0.483 | 0.496 | 0.458 | 0.441 | 0.420 | 0.414 |
| TNR | 0.963 | 0.927 | 0.818 | 0.846 | 0.517 | 0.504 | 0.542 | 0.559 | 0.580 | 0.586 |
| FNR | 0.047 | 0.054 | 0.120 | 0.095 | 0.460 | 0.466 | 0.310 | 0.394 | 0.414 | 0.457 |

## 5.2 Main Results: Diffusion MIA Fails on Real-world Setups

We use five setups in CopyMark to evaluate state-of-the-art MIA method on diffusion models and show the result in Table 5. Unfortunately, while they perform consistently with the result in the original papers on the previous setups, **all MIA methods fail on our new real-world setups**.

**MIAs perform consistently with results in the original papers** We compare our results on setup **(a)** and **(b)** to the results in the original paper of MIA methods to cross-validate the correctness of our implementation. In the original paper of PFAMI (Fu et al., 2023), its AUC on setup **(a)** is 0.961, while our result is 0.9172. Our result is slightly lower than the original result. However, both result are in the same level. In the original paper of SecMI (Duan et al., 2023) and PIA (Kong et al., 2023), the TPRs@1%FPR are 0.1858 and 0.198, and the AUCs are 0.701 and 0.739, respectively. Our results are 0.3120 and 0.2888 for the TPR@1%FPR and 0.7617 and 0.6991 for the AUC, which are slightly better. The slight difference between our results and the original results are acceptable and should be attributed to the different in random seed and dataset sampling. The consistent performance of baselines on setup **(a)** and **(b)** validates the correctness of our implementation.

**Loss-based MIAs fail on real-world setups** On setup **(c)**, **(d)**, and **(e)**, however, loss-based MIA methods (SecMI, PIA, and PFAMI) fail. Their TPRs@1%FPR and TPRs@0.1%FPR are close to the FPR threshold, indicating their disability in distinguishing even a few members from non-members.

**Classifier-based MIAs fail on real-world setups** Compared to loss-based MIAs, classifier-based MIAs ($GSA_1$ and $GSA_2$) perform better on three real-world setups. $GSA_1$ and $GSA_2$ always succeed in separating members and non-members in the validation dataset because they exploit the classifier trained on the same dataset. However, when transferring the classifier to the test dataset, the performance degrades significantly. First, the TPRs drop to the range of $0.55 - 0.89$. Second, the FPRs rise dramatically to the range of $0.10 - 0.43$, much higher than the original FPR of the threshold. We use red number to note the test FPRs higher than the validation FPRs. In other word, classifier-based MIAs tend to classify $10\% - 40\%$ of the non-members as members. Therefore, they cannot be trustworthy evidence of diffusion model membership either.

**Blind baseline beats loss-based MIAs** It is noticeable that the Blind baseline, based on a ConvNext classifier, yields competitive performance. On setups **(a)** and **(b)**, it outperforms all MIA methods.

Table 5: Benchmark results of MIA methods on CopyMark. Red means FPR on the test set is higher than FPR upper-bound $X\%$ on the evaluation set.

**(a)** Latent Diffusion Model (CelebA-HQ / FFHQ)

| | Evaluation Set | | | Test Set | | | |
|---|---|---|---|---|---|---|---|
| | TPR@1%FPR | TPR@0.1%FPR | AUC | $TPR_{1\%}$ | $FPR_{1\%}$ | $TPR_{0.1\%}$ | $FPR_{0.1\%}$ |
| SecMI | 0.0728 | 0.0028 | 0.6131 | 0.0696 | 0.0084 | 0.0012 | 0.0004 |
| PIA | 0.0228 | 0.0016 | 0.6250 | 0.0252 | 0.0100 | 0.0004 | 0.0004 |
| PFAMI | 0.4988 | 0.2036 | 0.9172 | 0.5016 | 0.0192 | 0.1916 | 0.0012 |
| $GSA_1$ | **1.0000** | **1.0000** | **1.0000** | 0.9516 | 0.0120 | 0.9516 | 0.0120 |
| $GSA_2$ | **1.0000** | **1.0000** | **1.0000** | 0.9492 | 0.0132 | 0.9492 | 0.0132 |
| Blind | **1.0000** | **1.0000** | **1.0000** | **0.9932** | 0.0092 | **0.9932** | 0.0092 |

**(b)** Stable Diffusion v1.5 (LAION Aesthetic V2 5+ / MS-COCO2017)

| | Evaluation Set | | | Test Set | | | |
|---|---|---|---|---|---|---|---|
| | TPR@1%FPR | TPR@0.1%FPR | AUC | $TPR_{1\%}$ | $FPR_{1\%}$ | $TPR_{0.1\%}$ | $FPR_{0.1\%}$ |
| SecMI | 0.2888 | 0.1364 | 0.6991 | 0.3096 | 0.0084 | 0.1508 | 0 |
| PIA | 0.3120 | 0.1776 | 0.7617 | 0.3420 | 0.0112 | 0.1912 | 0 |
| PFAMI | 0.2124 | 0.1048 | 0.5870 | 0.2068 | 0.0072 | 0.1004 | 0 |
| $GSA_1$ | **1.0000** | **1.0000** | **1.0000** | 0.8592 | 0.0968 | 0.8592 | 0.0968 |
| $GSA_2$ | **1.0000** | **1.0000** | **1.0000** | 0.8556 | 0.0844 | 0.8556 | 0.0844 |
| Blind | **1.0000** | **1.0000** | **1.0000** | **0.9004** | 0.1156 | **0.9004** | 0.1156 |

**(c)** Stable Diffusion v1.5 (LAION-Members / LAION-Non-members)

| | Evaluation Set | | | Test Set | | | |
|---|---|---|---|---|---|---|---|
| | TPR@1%FPR | TPR@0.1%FPR | AUC | $TPR_{1\%}$ | $FPR_{1\%}$ | $TPR_{0.1\%}$ | $FPR_{0.1\%}$ |
| SecMI | 0.0128 | 0.0020 | 0.5231 | 0.0108 | 0.0088 | 0.0004 | 0.0004 |
| PIA | 0.0128 | 0.0020 | 0.5352 | 0.0124 | 0.0088 | 0.0004 | 0.0004 |
| PFAMI | 0.0156 | 0.0032 | 0.5101 | 0.0104 | 0.0108 | 0.0016 | 0.0020 |
| $GSA_1$ | **1.0000** | **1.0000** | **1.0000** | **0.7016** | 0.2704 | 0.5608 | 0.4184 |
| $GSA_2$ | **1.0000** | **1.0000** | **1.0000** | 0.6680 | 0.2736 | **0.5780** | 0.4056 |
| Blind | 0.9968 | 0.9520 | 0.6848 | 0.4592 | 0.3938 | 0.1432 | 0.1124 |

**(d)** CommonCanvas-XL-C (CommonCatalog-CC-BY / MS-COCO2017)

| | Evaluation Set | | | Test Set | | | |
|---|---|---|---|---|---|---|---|
| | TPR@1%FPR | TPR@0.1%FPR | AUC | $TPR_{1\%}$ | $FPR_{1\%}$ | $TPR_{0.1\%}$ | $FPR_{0.1\%}$ |
| SecMI | 0.0092 | 0.0004 | 0.5000 | 0.0080 | 0.0060 | 0 | 0 |
| PIA | 0.0124 | 0.0004 | 0.5184 | 0.0172 | 0.0084 | 0 | 0 |
| PFAMI | 0.0124 | 0.0004 | 0.5034 | 0.0208 | 0.0132 | 0 | 0 |
| $GSA_1$ | **1.0000** | **1.0000** | **1.0000** | **0.8912** | 0.3132 | **0.8912** | 0.3132 |
| $GSA_2$ | **1.0000** | **1.0000** | **1.0000** | 0.8880 | 0.1052 | 0.8880 | 0.1052 |
| Blind | 0.9984 | 0.9568 | 0.9998 | 0.8804 | 0.1564 | 0.7348 | 0.0624 |

**(e)** Kohaku-XL-Epsilon (HakuBooru-Members / HakuBooru-Non-members)

| | Evaluation Set | | | Test Set | | | |
|---|---|---|---|---|---|---|---|
| | TPR@1%FPR | TPR@0.1%FPR | AUC | $TPR_{1\%}$ | $FPR_{1\%}$ | $TPR_{0.1\%}$ | $FPR_{0.1\%}$ |
| SecMI | 0.0116 | 0.0169 | 0.5008 | 0.0116 | 0.0000 | 0.0136 | 0 |
| PIA | 0.0076 | 0 | 0.5051 | 0.0096 | 0.0128 | 0 | 0 |
| PFAMI | 0.0104 | 0.0008 | 0.4979 | 0 | 0 | 0 | 0 |
| $GSA_1$ | **1.0000** | **1.0000** | **1.0000** | **0.5668** | 0.4192 | **0.5668** | 0.4192 |
| $GSA_2$ | **1.0000** | **1.0000** | **1.0000** | 0.5536 | 0.4292 | 0.5536 | 0.4292 |
| Blind | 0.9736 | 0.9584 | 0.9997 | 0.4204 | 0.3064 | 0.3528 | 0.2444 |

This again indicates the defect of these previous evaluation setups because they can be totally covered without accessing to the diffusion model. On our real-world setups **(c)**, **(d)**, and **(e)**, the blind baseline beats loss-based MIAs with a similar performance to that of classifier-based MIAs. This proves that classifier-based methods have some practical value. It also shows that our setups require the methods to depend more on the membership rather than the distribution shift between members and non-members, which is our superiority. We believe that all future MIA methods should be compared to this blind baseline in the evaluation.

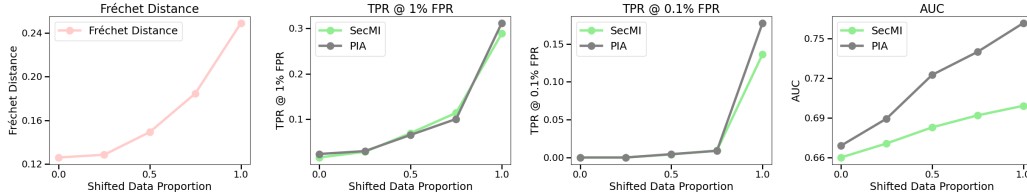

Figure 1: **Left:** Frechet distances increase with the proportion of shifted data in the non-member dataset. **Right×3:** Three performance metrics also increase with the proportion of shifted data in the non-member dataset. This shows that we could manipulate the result of diffusion MIA by changing the component of non-member datasets, which is expected to irrelevant to the result.

**Loss-based MIAs generalize better than classifier-based MIAs** Throughout all five setups, we notice that loss-based MIAs have good generalizability, that they perform consistently on test datasets as on validation datasets. They seldom yield a test FPR higher than the FPR of validation threshold. In contrast, classifier-based MIAs suffer from the performance gap between validation datasets and test datasets. This difference in generalizability is straight-forward to understand: Classifier-based MIAs depends on a neural network that tends to over-fit the features of data points to achieve the perfect performance on the validation dataset. This over-fitting results in performance degradation on the test dataset. To eliminate this problem, we must further refine the feature selection.

### 5.3 HOW DO DATASET SHIFTS IMPACT MIA PERFORMANCE?

We further investigate how different levels of dataset shifts impact the performance of diffusion MIA. To this end, we focus on Stable Diffusion 1.5, because it has two different non-member datasets in our benchmark, where LAION-Non-member does not have dataset shifts and COCO2017-Val does (the reference member dataset is our LAION dataset). This provides possibilities to construct a series of non-member datasets with different levels of dataset shifts by interpolation. We then construct five non-member datasets by mixing data from above two non-member datasets with proportions of 0%:100%, 25%:75%, 50%:50%, 75%:25%, and 100%:0%. We calculate their Frechet distance and then evaluate two loss-based MIA methods, SecMI and PIA, on these non-member datasets and the fixed LAION member dataset. Figure 1 demonstrates the result. The Frechet distance increases with the increase of the proportion of COCO2017-Val, the non-member dataset with dataset shifts. Consistent with our expectation, the performance of two MIA methods have monotonously positive correlation with the the dataset shifts on different levels of dataset shifts. In other words, they perform better when there is a more serious dataset shift.

Notably, the setup of 100%:0% is the most widely used MIA benchmark (setup **b**). We show that one can manipulate the result of MIA evaluation easily by only changing non-members, which is supposed to be irrelevant to the result. This indicates that current MIA evaluation is unreliable.

## 6 DISCUSSION

Due to the page limit, we omit our discussion to the appendix. Specifically, we first analyze why current MIA methods all fail on the real-world benchmark and give a brief insight on how to overcome the current bottlenecks of diffusion MIA (Appendix A.3.1). Then, we discuss the impact of our works on the practice in copyright lawsuits (Appendix A.3.2).

## 7 CONCLUSION

In this paper, we reveal two defects in the previous evaluation of membership inference attacks (MIAs) on diffusion models: over-training and dataset shifts, which result in overestimate of MIAs' performance. To overcome these defects, we propose CopyMark, the first unified benchmark for MIAs on diffusion models. Our choice of models and datasets keeps CopyMark away from over-training and dataset shifts. We evaluate existing MIA methods with CopyMark and find that current MIAs on diffusion models fail in real-world scenarios of MIA. We alert that MIAs on diffusion models are not trustworthy tool to provide evidence for unauthorized data usage in diffusion models.

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

## A  APPENDIX

### A.1  IMPLEMENTATION DETAILS

**diffusers** diffusers abstracts inference workflows of diffusion models as *pipelines*. A pipeline loads and manages *Modules* of diffusion models. Then, it takes inputs and returns outputs. For example, StableDiffusionImage2ImagePipeline takes images and text prompts as inputs and uses modules of Stable Diffusion to sample output images with SDEdit (Meng et al., 2021). Usually, state-of-the-art diffusion models consist of three modules: U-Net (UNet2DModel), VAE (AutoencoderKL), and text encoder (CLIPTextModel (Radford et al., 2021)). One diffusion model only have one set of modules. However, it may have several different pipelines. For example, StableDiffusion-Img2ImgPipeline (Meng et al., 2021), StableDiffusionInstructPix2PixPipeline (Brooks et al., 2023), and StableDiffusionGLIGENPipeline (Li et al., 2023) are all pipelines of Stable Diffusion v1.5.

**Baselines** We first introduce the idea of our baselines as follows:

- **SecMI** (Duan et al., 2023): A loss-based method. SecMI uses a parameterized forward step $p_\theta(x_t|x_{t-1}, x_0)$ to predict $x_t$ from $x_{t-1}$. Then, it applies a reverse step $p_\theta(x_t|x_{t-1}, x_0)$ to predict $\widetilde{x}_{t-1}$. The score is given by the $l_2$ distance between $x_{t-1}$ and $\widetilde{x}_{t-1}$, that diffusion models should better predict $x_{t-1}$ for member images. SecMI use the distance at $t = 50$, while we also try a variant that uses the distance at different $t$, termed by SecMI++.

- **PIA** (Kong et al., 2023): A loss-based method. PIA distinguishes member data by checking how diffusion models denoise the $x_t$ with the same noise $\epsilon_0$. The score is given by the loss computed over different timesteps $t$.

- **PFAMI** (Fu et al., 2023): A loss-based method. PFAMI exploits the loss fluctuation in the image neighborhood. The neighborhood of image $x$ refers to images that share similar contents with $x$ and is constructed by cropping $x$. It compares the loss of $x$ with that of its neighborhood. It assumes that the loss of non-member images approximates those of their neighborhood images, while the loss of member images should be a distinct local minimum among those of the neighborhood images.

- **GSA** (Pang et al., 2023): The first gradient-based method. GSA aggregates gradients on modules in diffusion models over different timesteps $t$ and uses the $l_2$ norm of these gradients as features. Then, it trains an XGBoost (Chen & Guestrin, 2016) binary classifier to discriminate features from member data and those from non-member data. The score is given by the classifier. GSA has two variants, termed by $GSA_1$ and $GSA_2$.

All baselines are implemented based on their official open-sourced implementation. For SecMI, we use DDIM (Song et al., 2020a) as the sampling method with $\eta = 0$ and pick the score at $t = 50$ as advised by the official implementation [3]. For PIA, we follow the official implementation [4] to compute losses at $t \in \{0, 10, 20, ..., 480\}$. For PFAMI, we follow the original setup in the official implementation [5] to set the neighbor number $N$ as 10, the attacking number $M$ as 1, and the interval of perturbation strengths as $[0.75, 0.9]$. For the above methods, we separate the threshold interval $[\tau_{min}, \tau_{max}]$ in Algorithm 1 into 10,000 sub-intervals and pick the corresponding 10,000 lower-bounds for optimal threshold searching. For $GSA_1$ and $GSA_2$, we use the default setup [6] to compute losses ($GSA_1$) or gradients ($GSA_2$) over different modules at $t \in \{0, 50, 100, ..., 1000\}$. An XGBoost (Chen & Guestrin, 2016) classifier with 200 estimators is trained on the gradients to distinguish member images, following the original implementation. The threshold of $GSA_1$ and $GSA_2$ is fixed to 0.5 because the XGBoost classifier outputs binary scores of $\{0, 1\}$. For all methods, we use the seed function from the official implementation of SecMI [7] that fixes the seed as 1 to make the result deterministic.

**Sanity Check** Setup **(a)** and **(b)** are mirroring setups of previous evaluation setups (Duan et al., 2023; Fu et al., 2023). Hence, we do not repeat their sanity checks. The sanity check of setup **(c)** has been done by Dubiński et al. (2024). For Setup **(d)**, CommonCanvas-XL-C uses MS-COCO2017 as its validation set for generation performance (Gokaslan et al., 2023). This indicates that MS-COCO2017 is held out of the training dataset of CommonCanvas-XL-C. In Setup **(e)**, every data point in HakuBooru dataset has a unique ID, which the author of Kohaku-XL-Epsilon used to select the training dataset from the whole dataset. We follow the instruction in the homepage [8] to randomly pick images from the ID range of the training dataset as members and images out of this ID range as non-members. Hence, there should be no overlap between members and non-members.

**Computational resources** All experiments are finished on $2\times$ NVIDIA A100 80GB GPUs.

## A.2 ADDITIONAL RELATED WORKS

Unauthorized data usage of diffusion models have been a crucial topic that raises increasing attention (Franceschelli & Musolesi, 2022; Sag, 2023; Samuelson, 2023). One popular approach to relieve the concern of unauthorized data usage in diffusion models is to add adversarial (Salman et al., 2023; Shan et al., 2023a; Liang et al., 2023; Liang & Wu, 2023; Van Le et al., 2023; Shan et al., 2023b; Xue et al., 2023) or copyright watermarks (Cui et al., 2023; Zhu et al., 2024) to images. These watermarks either resist diffusion models from training on the image or introduce copyright information to diffusion models trained on the image. However, recent research questions the effectiveness of these watermarks that they might be failed easily (Zhao et al., 2023). Another recipe is to erase or *unlearn*

---

[3] https://github.com/jinhaoduan/SecMI-LDM
[4] https://github.com/kong13661/PIA
[5] https://anonymous.4open.science/r/MIA-Gen-5F40/
[6] https://github.com/py85252876/GSA
[7] https://github.com/jinhaoduan/SecMI-LDM/blob/secmi-ldm/src/mia/secmi.py
[8] https://huggingface.co/KBlueLeaf/Kohaku-XL-Epsilon

Table 6: Complexity analysis and running time for MIA methods. Query (FP) and Query (BP) are the number of forward propagations and back propagation the method conducts per image. Running time is given based on the whole experiments of 2500 images on three models: Latent Diffusion, Stable Diffusion v1.5, and CommonCanvas-XL-C. Experiments are conducted on an NVIDIA A100 GPU.

| | Query (FP) | Query (BP) | Time (LDM) | Time (SD) | Time (CC-XL-C) |
|---|---|---|---|---|---|
| SecMI | 100 | 0 | 2339 | 8965 | 16280 |
| PIA | 100 | 0 | 2804 | 9331 | 20397 |
| PFAMI | 1100 | 0 | 47171 | 40787 | 47909 |
| $GSA_1$ | 20 | 1 | 5886 | 12787 | 18146 |
| $GSA_2$ | 20 | 20 | 6729 | 14143 | 26771 |

the copyright data in the diffusion model (Gandikota et al., 2023; Zhang et al., 2023a; Fan et al., 2023; Wu et al., 2024; Zhang et al., 2024). Nevertheless, Zhang et al. (2023b) shows that current machine unlearning on diffusion models can be bypassed by soft prompting and other fine-tuning methods. As protective and post-training refining methods are faced with questioning, we highlight the potential to directly detect copyright data in the training dataset of diffusion models, by which we help the calling for copyright protection beyond technical methods.

We are the first to validate and alert the defect in the evaluation of MIAs on diffusion models. Some works reported similar risks in MIAs on Large Language Models (Das et al., 2024; Maini et al., 2024). Liu & He (2024) shows that most vision dataset pairs could be classified by a simple classifier, which inspires our investigation to the previous evaluation of MIAs on diffusion models. Our unified benchmark for MIAs on diffusion models covers the dataset for real-world evaluation of MIAs in Dubiński et al. (2024). However, they do not implement any MIA methods on their dataset.

### A.3 DISCUSSION

#### A.3.1 WHY DO CURRENT DIFFUSION MIA METHODS FAIL?

Two kinds of MIAs, loss-based MIAs (Duan et al., 2023; Kong et al., 2023; Fu et al., 2023) and classifier-based MIAs (Pang et al., 2023), fail for different reasons:

**Loss-based MIAs:** These methods are built on the hypothesis that training losses of members are lower than those of non-members. However, pre-trained diffusion models were trained on one data point for one iteration. Hence, the difference between member training losses and non-member training losses are smaller. Also, the training loss depends on the Gaussian noise added to the clean data and the time step. While current loss-based MIA simply use randomly sampled noise and time steps to calculate losses, it is difficult to accurately locate the noise and the time step where the loss is minimized. That is the reason why the calculated loss is not a good measure for membership. To overcome this, we should take into consideration the loss dependency on noise and time step. This will make us better locate the exact loss being optimized and then distinguish members.

**Classifier-based MIAs:** Classifier-based MIA methods are built on the assumption that image representations of members and non-members in the diffusion models are distinguishable. However, they suffer the same problem with loss-based methods, that the image representation is also time-step-dependent and noise-dependent. This means that classifier-based MIA methods also need to detect the exact noise and time step used in that one training epoch for the specific data point.

Generally, both current loss-based methods and classifier-based methods neglect the fact that any features in diffusion models have high dependency on the time step and the noise. Without modeling this dependency, it would be always difficult to determine the membership.

#### A.3.2 ARE MIAS POTENTIAL EVIDENCE OF UNAUTHORIZED DATA USAGE IN AI LAWSUITS?

Recent progress in AI copyright lawsuits (Andersen, 2023; Zhang, 2024) indicates the necessity of evidence of unauthorized data usage in pre-trained diffusion models. Specifically, the plaintiff claims that pre-trained diffusion models copy their copyright images without authorization and expects to get evidence of this copying by MIAs on diffusion models. According to copyright laws (U.S.

Copyright Office, 2021; European Parliament and Council, 2001), however, the proof of copying requires showing *substantial similarity* between the defendant's work and original elements of the plaintiff's work. In the context of AI copyright lawsuits, this means that the plaintiff must provide images generated by pre-trained diffusion models with content similarities to their copyright images. Unfortunately, today's MIAs could only give binary membership indicators as outputs. Moreover, as shown in this paper, current MIAs on diffusion models are not reliable tools to even indicate membership. Hence, it is non-realistic to make use of MIAs as evidence in AI copyright lawsuits.

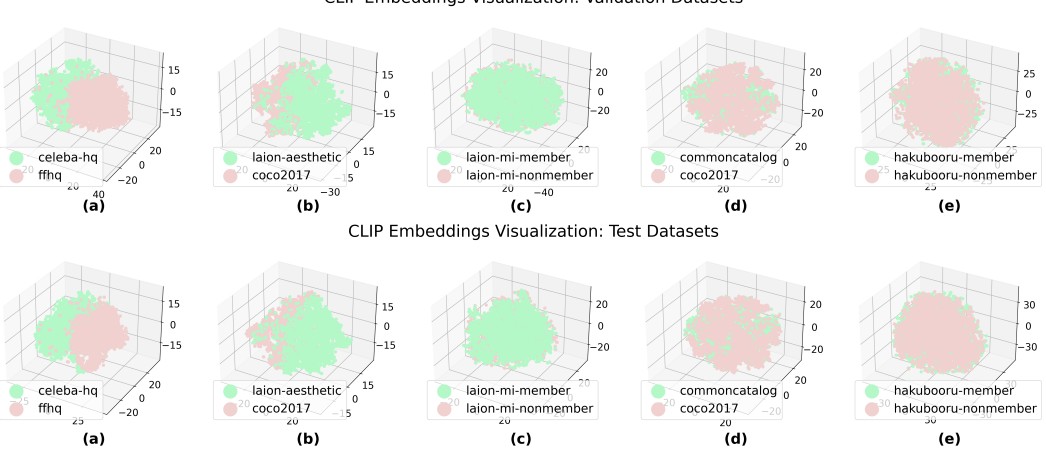

Figure 2: Visualizing compressed CLIP embeddings of members and non-members of our 5 evaluation setups. **(a)** and **(b)**, two defective setups, have their members and non-members markably distinguishable. In contrast, members and non-members of other three new setups, **(c)**, **(d)** and **(e)**, cannot be well separated in the CLIP embedding space.

## A.4 CODEBASE OF COPYMARK ON DIFFUSERS

The codebase of CopyMark on diffusers mainly consists of three parts: diffusers pipeline, data and evaluation utilities, and training scripts.

**diffusers pipelines** We implements every MIA method as a pipeline in diffusers. Pipelines of one diffusion model (e.g. Stable Diffusion) are consistently inherited from the model's text-to-image pipeline (`StableDiffusionPipeline` for Stable Diffusion and `StableDiffusionXLPipeline` for SDXL) or the unconditional generation pipeline (`DiffusionPipeline` for Latent Diffusion Models). These pipelines load modules with the unified module loading API of diffusers. They differs from the parent pipeline only by the `__call__()` function. We modify their `__call__()` to take images as inputs and return the result as outputs. We list all pipelines implemented as follows:

```
SecMILatentDiffusionPipeline
SecMIStableDiffusionPipeline
SecMIStableDiffusionXLPipeline
PIALatentDiffusionPipeline
PIAStableDiffusionPipeline
PIAStableDiffusionXLPipeline
PFAMIMILatentDiffusionPipeline
PFAMIStableDiffusionPipeline
PFAMIStableDiffusionXLPipeline
GSALatentDiffusionPipeline
GSAStableDiffusionPipeline
GSAStableDiffusionXLPipeline
```

SecMI & SecMI++ and $GSA_1$ & $GSA_2$ share one pipeline respectively with different arguments.

**Data and evaluation utilities** Since all pipelines take images as inputs and return scores as outputs, we use a set of unified utilities to load the images and optimize the threshold from the scores.

**Training scripts** The training script of one MIA method assembles the diffusers pipeline and the utilities. It first loads images and text prompts with the data utility. Then, it employs the diffusers pipeline to calculate scores. Finally, it uses the evaluation utility to optimize the threshold according to the scores and evaluate the threshold.

