# OpenReview forum: "Real-World Benchmarks Make Membership Inference Attacks Fail on Diffusion Models"
_ICLR.cc/2025/Conference — Submitted to ICLR 2025_

### Official Review · Reviewer_y1k9 · 2024-10-29

**Soundness:** 3
**Presentation:** 3
**Contribution:** 2
**Rating:** 5
**Confidence:** 3

**Summary:**

This paper investigates the evaluation of state-of-the-art membership inference attacks (MIAs) on diffusion models in real-world scenarios. Specifically, it highlights flaws in current MIA evaluations, where over-training and dataset shifts lead to overestimated performance of the membership detection. To address this, the paper introduces a unified benchmark for MIAs on diffusion models, named CopyMark, which is built without over-training, using non-shifted datasets and blind testing. The experiments cover the recent loss-based MIA methods and classifier-based MIA methods, conducted on both defective setups and real-world setups. The results reveal that existing MIAs perform poorly on diffusion models in realistic scenarios.

**Strengths:**

1. The paper is well-written and easy to follow. It explains the flaws in existing MIA evaluations, i.e., over-training and dataset shifts, and is structured to understand these two problems through quantitative and qualitative analyses.

2. This paper makes valuable thoughts about the limitations of current MIA evaluations on diffusion models. The significance of the proposed realistic evaluation for MIA is substantial, particularly in the context of AI copyright lawsuits and data privacy.

**Weaknesses:**

1. The originality of these two flaws, i.e., over-training and dataset shifts, remains a concern. Similar concepts like over-fitting and distribution shifts have been discussed in previous works (Carlini et al., 2022; Maini et al., 2024) on traditional deep learning models and large language models. This paper may potentially adapt the MIA setting to diffusion models while providing more assessments.

2. Although the paper assesses existing MIA methods on diffusion models, it does not explore possible adjustments to improve MIA performance on CopyMark. For example, how to address the challenges identified on existing loss-based and classifier-based MIA methods and how to achieve better results under realistic scenarios.

3. The evaluation may lack comprehensiveness as a benchmark, as the experiments are limited to loss-based and classifier-based MIA methods on diffusion models. Other types of MIAs, such as likelihood-based MIAs (Hu & Pang, 2023) and MIAs using Quantile Regression (Tang et al., 2024), are not included.

Carlini et al. Membership inference attacks from first principles. 2022 IEEE Symposium on Security and Privacy. 2022.

Maini et al. LLM Dataset Inference: Did you train on my dataset? arXiv preprint arXiv:2406.06443. 2024.

Hu & Pang. Loss and likelihood based membership inference of diffusion models. In International Conference on Information Security. 2023.

Tang et al. Membership inference attacks on diffusion models via quantile regression. International Conference on Machine Learning. 2024.

**Questions:**

1. How do the issues of over-training and dataset shifts differ between diffusion models and traditional deep learning models or large language models? Will the proposed realistic scenarios similarly reduce MIA effectiveness on these other model types?

2. How do MIA methods based on likelihood and quantile regression perform on diffusion models in the proposed realistic scenarios? Will their performance also see a significant reduction?

3. Minor point: a typo “randomlyy” in the third paragraph of section 4.3.

---

> ### Author Response · Authors · 2024-11-21
>
> We thank the insightful review and would like to address the issue by points:
>
> [W1 & Q1] (difference from overfitting & our contribution) We agree that over-fitting and distribution shifts have been discussed by previous or concurrent works. However, there are differences between over-training of diffusion models and traditional over-fitting. Over-fitting in traditional models could be measured by train-test gaps [C] and is supposed to be a bad phenomena. However, as discussed in L155-L158, over-training (for hundreds of epochs) is necessary to achieve the best FID for small diffusion models. For example, the default training epochs of DDPM on CIFAR-10 is 2048 (=800k*128 / 50k) epochs. Using these default small models is the reason why current MIA evaluation tends to suffer from over-training. However, as shown by our experiments, MIA succeeding on over-trained models may have nothing to do with large-scale models without over-training, e.g. Stable Diffusion, while the latter constitutes the real world concern of MIA. This dilemma is kind of unique for diffusion models. We have mentioned this in our updated new draft in Section 3.
>
> Furthermore, the contribution of this work is more on rectifying current academic practice, that we aim at stopping the wrong trend of depending on flawful evaluation. As mentioned by the reviewer, there are lots of works discussing the over-fitting and distribution shifts in MIA of other models. However, new research in diffusion MIA seems to neglect these discussions and continues to use evaluation with over-training and dataset shifts. For example, one of our concurrent works (link: https://openreview.net/forum?id=LRSspInlN5) still uses DDPM+CIFAR10, DDPM+CIFAR100, and our setup (b) (flawful) as benchmarks. Hence, we believe it is significant and timely to explicitly alert the flaw of current MIA evaluation and call for stopping depending on it. In addition, we provide three prepared and plug-and-play benchmarks, our setup (c) (d) and (e), for realistic MIA evaluation. We do this by searching for all possible large-scale diffusion models with accessible and no-shift members and non-members. This helps future diffusion MIA research adapt swiftly to the real-world setup. We believe both of these two contributions are not trivial.
>
> [W2] (possible adjustments to improve MIA) To provide insights for future adjustments, we have conducted new experiments to demonstrate the correlation between quantified dataset shifts and MIA performance, which validates that with only dataset shifts we can make the MIA fail (Section 5.3). Also, we have updated the analysis of the reason for MIA failure and given brief insights on potential improvements (Appendix A.3.1). To briefly summarize the idea, due to the fact that we only train one step on one specific noise and time step, we cannot distinguish members based on losses with randomly sampled time steps and noises. Instead, we may need to locate the exact time step and noise that the model used for THAT training step.
>
> However, as discussed above, our main goal is to alert the fatal flaw in current diffusion MIA evaluation and provide a realistic benchmark. Designing novel methods is then out-of-the-scope of our primary focus. Hence, we would like to leave this for future work.
>
> [W3 & Q2] (new baselines) We thank the reviewer for the advice and feel sorry for missing these baselines. We have included these works in our references and are currently running experiments on them. However, it is notable that likelihood-based methods [A] perform similarly to loss-based methods and are not considered as baselines in recent MIA research. Additionally, [B] is a quantiled version of GSA in our baselines, which mainly focuses on improving efficiency. Hence, we believe these two baselines will not be exceptions for our conclusion on the diffusion MIA’s failure on real-world benchmarks.
>
> [Q3] (typos) We thank the reviewer and have fixed it in our updated new draft.
>
> Again, we thank the reviewer for the insightful review. If you have further questions, feel free to contact us.
>
> References:
>
> [A] Hu & Pang. Loss and likelihood based membership inference of diffusion models. In International Conference on Information Security. 2023.
>
> [B] Tang et al. Membership inference attacks on diffusion models via quantile regression. International Conference on Machine Learning. 2024.
>
> [C] Carlini et al. Membership inference attacks from first principles. 2022 IEEE Symposium on Security and Privacy. 2022.

---

> ### Author Response · Authors · 2024-12-02
> **Kindly request for further discussions**
>
> Dear Reviewer,
>
> We highly appreciate your constructive reviews that raised the concerns on the contribution of our paper. As we mentioned in the rebuttal, the main contribution of this paper is to alert a wrong trend of the current MIA research on diffusion models. While MIA benchmarks are not necessary to be defective, all existing benchmarks suffer from the two defects we presented and are continuously used in evaluating new methods. Hence, we believe it is necessary to point out the defects and provide a new fair benchmark. We believe our work is significant for the practice of MIA research in belief of the necessity of a fair benchmark. As the discussion phase is about to close, we will sincerely appreciate it if you can engage the discussion and provide further advice on our paper.

---

### Official Review · Reviewer_iXFF · 2024-11-03

**Soundness:** 3
**Presentation:** 3
**Contribution:** 3
**Rating:** 6
**Confidence:** 3

**Summary:**

The paper presents a novel approach to assessing MIA on diffusion models by introducing a new benchmark called CopyMark. This benchmark aims to provide a realistic and unbiased environment for testing the effectiveness of MIAs against these models. The study underscores the potential overestimation of MIA effectiveness due to biased experimental setups in previous research and argues for a more nuanced understanding and evaluation of MIAs in practical applications. The paper pinpoints that current MIAs on diffusion models are not trustworthy tool to provide evidence for unauthorized data usage in diffusion models.

**Strengths:**

1. Good and significant topic. The paper identifies a critical gap in the evaluation of MIAs, offering a novel approach to benchmarking that could reshape how these attacks are studied, and providing valuable insights that can influence the future research.
2. Comprehensive experiment. The experiments conducted are extensive, providing evidence that challenges the overestimation of MIA effectiveness on diffusion models.

**Weaknesses:**

1. Lack of discussion. The discussion on the practical implications of the findings is somewhat superficial and lacks depth in Section 6, particularly in how these results could influence real-world security strategies.

**Questions:**

The paper aims to construct a real-world benchmark, pinpointing the current limitation of MIA setups, specifically the unknown distribution of members and non-members in real-world MIAs. It is reasonable that a newly proposed benchmark can cause current methods to yield poor performance. However, I find the discussion lacking in adequately demonstrating how this benchmark accurately reflects real-world settings from my perspective. In my opinion, additional evidence and a more thorough discussion would strengthen this aspect.

In the evaluation setup part, the paper mentions that (d) has a slight data shift but is more minor than other settings. Can you provide further insight into how minor dataset shifts were quantified and their potential impact on the validity of MIA results? It would be beneficial to have a more detailed analysis of how significant these shifts need to be in impacting the effectiveness of MIAs. What thresholds for dataset similarity were considered, and how were they determined?

The paper demonstrates that current MIAs are less effective under realistic conditions on diffusion models. How do you envision these findings being applicable to other types of generative models? Are there specific characteristics of diffusion models that may limit the generalizability of the results? A discussion on this could clarify potential broader applications of your findings.

This paper conducts a comprehensive experiment and concludes that the current MIAs on diffusion models do not perform well in real-world scenarios. However, I think the discussion part is relatively superficial and requires a deeper analysis based on the experimental results. Can you provide more implications and extend the discussion to promote future research?

---

> ### Author Response · Authors · 2024-11-21
>
> We thank the insightful review and would like to address the issue by points:
>
>
> [Q1] (Why our benchmark is realistic) Our benchmark is more realistic than existing benchmarks, because we use models without over-training and non-member datasets with minor or without shifts to member datasets. We would like to summarize the reasons as follows:
>
>
> Dataset shifts: Membership inference is supposed to only depend on membership. If we use a cat datasets to train a model and use a dog dataset as non-members for evaluation, an image classifier can easily separate these two datasets. However, this is not membership inference, and will immediately fail when we use another hold-out cat dataset as non-members. According to our new experiments in Section 5.1, our datasets do have smaller dataset shifts. Also, Section 5.3 shows that by only tuning the dataset shifts, we can manipulate the evaluation result. This means that current evaluation with big dataset shifts is not fair and realistic.
>
>
> Over-training: over-training diffusion models for hundreds of epochs on small datasets make the loss of members conspicuously lower than that of non-members. However, real-world security and privacy scenarios mostly involve large-scale diffusion models that are only trained for one epoch on the training dataset. MIA succeeding on over-training models may have nothing to do with these models in the real-world application. Our benchmark uses pre-trained models with only one epoch, which is the minimum training epoch, thus getting rid of over-training.
>
>
> [Q2 & W1] (in-depth analysis on dataset shift’s impact) We thank the reviewer for the advice and have conducted new experiments to quantitatively analyze how dataset shifts have impact on the MIA performance. Basically, our experiments have two parts:
>
>
> - Quantifying dataset shifts (Section 5.1). We calculate three distance metrics between member and non-member datasets in our benchmark: normalized Wasserstein distance (NWD), Fréchet Distance (FD), and Mahalanobis Distance (MD). All distances are calculated based on CLIP-large. Among all setups covered, setup (a) and (b) in our benchmark suffer much bigger distances between members and non-members, for example, FDs of 0.32 and 0.24. This validates our conclusion that there are valid shifts between their members and non-members. Therefore, MIA methods could separate these two datasets according to the semantics rather than the membership, which raises the dataset shift concern in the paper. Setup (d) has medium distances between members and non-members, for example, a FD of 0.12. The distances are much smaller for Setup (c) and (e), for example, FDs < 0.10. This experiment quantitatively demonstrates the existence of dataset shifts and shows potential connections between dataset shifts and the MIA performance.
>
> - Relation between dataset shifts and MIA performance (Section 5.3). We construct a series of non-member datasets by mixing our two non-member datasets: COCO-val-2017 (with shifts to LAION) and LAION-MI (no shifts) with different proportions. We pick the proportions by 100% vs 0%, 75% vs 25%, 50% vs 50%, 25% vs 75%, and 0% vs 100%. We evaluate SecMI and PIA on these setups. The result in Figure 1 shows that there is positive correlation between the performance of these two MIA methods and the proportion of shifted non-members data. This shows that one can manipulate the result of MIA evaluation easily by only changing non-members, which is supposed to be irrelevant to the result and that current MIA evaluation is unreliable.
>
>
> [Q3] (generalization of our conclusion) We notice that there are existing works discussing the failure of MIA on other generative models, e.g. LLMs. However, MIA of LLMs seems to never fall into the hallucination of success as that of diffusion models do. This is because over-training is the default setup of training small diffusion models to achieve the best FID (L155-158 in our new draft). In other words, there is an intrinsic gap between MIA of small diffusion models and that of large-scale diffusion models in the real-world application. We believe this is unique for diffusion models. The main contribution of this work is also to rectify current academic practice in diffusion MIA, to stop the wrong trend of depending on evaluation pipelines far from real-world applications, and to provide a sound and realistic benchmark.

---

> ### Author Response · Authors · 2024-11-21
>
> [Q4] (implications to promote future research) To provide insights for future adjustments, we have updated the analysis of the reason for MIA failure and given brief insights on potential improvements (Appendix A.3.1). To briefly summarize the idea, due to the fact that we only train one step on one specific noise and time step, we cannot distinguish members based on losses with randomly sampled time steps and noises. Instead, we may need to locate the exact time step and noise that the model used for THAT training step.
>
> However, as discussed above, our main goal is to alert the fatal flaw in current diffusion MIA evaluation and provide a realistic benchmark. Designing novel methods is then out-of-the-scope of our primary focus. Hence, we would like to leave this for future work.
>
>
>  Again, we thank the reviewer for the insightful review. If you have further questions, feel free to contact us.

---

> ### Author Response · Authors · 2024-12-02
> **Kindly request for further discussion**
>
> Dear Reviewer,
>
> We highly appreciate your constructive reviews that raised the concerns on the lack of discussion. To address this concern, we have added discussions on the following topics. 1) We quantifying the impact of dataset shifts on MIA performance. The result shows that by only replacing the non-member dataset the MIA performance will suffer a significant drop. Hence, we need to consider the hardest non-member dataset as the real-world setup. 2) We discussion potential directions of future improvement in the area. We believe these two extra discussions can effectively address your concerns. As the discussion phase is about to close, we will sincerely appreciate it if you can engage the discussion and provide further advice on our paper.

---

### Official Review · Reviewer_eNEy · 2024-11-04

**Soundness:** 3
**Presentation:** 3
**Contribution:** 2
**Rating:** 5
**Confidence:** 3

**Summary:**

The paper proposed a simple but effective benchmark for evaluating the existing MIA’s performance on the pre-trained diffusion models for the data authorization problem. The authors first found that “overtraining” and “dataset shifts” are two major defects of the existing MIA methods. Then, to overcome the two challenges, the authors proposed a benchmark that incorporates five different experimental setups, where the last three avoids the dataset shifting problem by using members and non-members from the same distributions, and over-training problem by only considering pre-trained models training for 1 epoch.

**Strengths:**

- Presentation is good and easy to follow.
- The addressed problem is meaningful.

**Weaknesses:**

- I am confused about the upper part of Table 1. What do the “✅” and “❌” symbols represent in each entry? Additionally, are “Over-training” and “Shifted Datasets” considered issues in each experimental setup (e.g., is over-training a problem in the DDPM + CIFAR10 setup)? If so, why is over-training necessarily a problem for DDPM + CIFAR10? I believe this only holds when certain factors, like training epochs, are fixed as you reported in the common setting; otherwise, this claim seems overstated.

- Could the benchmark allow for more varied experimental setups—for instance, having no dataset shift but including over-training? A simple example could involve training a DDPM on the CIFAR10 training set and using the CIFAR10 test set as non-members, which would meet the no-shift criterion.

- Furthermore, the concept of “dataset shift” is somewhat unclear to me. The benchmark assumes there’s no distribution shift when two datasets come from the same source. I suggest the authors delve deeper into this by considering metrics to quantify dataset distance (distribution distance), such as the Wasserstein distance.

**Questions:**

See weakness.

---

> ### Author Response · Authors · 2024-11-21
>
> We thank the insightful review and would like to address the issue by points:
>
>
> [1] (Must over-training and dataset shifts occur in previous benchmarks?) We thank the reviewer for raising this meaningful question, for which our first draft does not provide a sound explanation. We would like to make a clarification as follows:
>
>
> First, ticks mean that the benchmark does have the over-training or dataset shifts while crosses mean it does not. There is a typo that "LDM + CelebA" should have both dataset shift and over-training in Table 1 (upper).
>
>
> Second, it is not a must that an arbitrary benchmark using the model and the dataset in Table 1 (upper) suffers from over-training and dataset shifts. Table is used to show the setups of all existing benchmarks with the two drawbacks. The model and the dataset are considered as parts of the shown setups.
>
>
> Third, however, while dataset shifts are easy to avoid (by using half of one dataset to train the model and another half as non-members), over-training is difficult to avoid for most small diffusion models. As mentioned in L155, training for hundreds of epochs is necessary to achieve the best FID for small datasets and diffusion models. This may be caused by the limit size of the dataset. Hence, most small diffusion models trained on academic datasets, including CIFAR-10, CelebA, and ImageNet, need over-training to converge. If we pick a small model without over-training, then the model could be unconverged and thus not qualified to benchmark MIA. That is why over-training is hard to avoid when using small diffusion models as MIA benchmarks.
>
>
> As a result, it is necessary to introduce large-scale diffusion models as real-world MIA benchmarks. These models enjoy the large size of the training dataset so that they could converge with only one epoch. Hence, MIA methods cannot rely on the memorization caused by over-training to easily distinguish members from non-members on these models. This is the main motivation of our work.
>
>
> [2] (Varied setups) As shown in Table 1, previous benchmarks have included such setups with over-training or dataset shifts. As discussed at L206-L208, these setups are meaningless. MIA benchmarks with dataset shifts can be dominated by image distribution classifiers, which will immediately fail on no-shift benchmarks, while those with over-training turn the problem into over-fitting detection. MIA methods succeeding on these benchmarks do not truly infer membership and cannot be used in real-world scenarios.
>
>
> The main contribution of our work is that we 1) alert a problematic trend of current MIA research that methods compete on above meaningless benchmarks and 2) provide a practical real-world benchmark for future research to compete on. While we have included two flawful setups a) and b) in CopyMark, we believe there is no need to include more, because this is not relevant to our two contributions.
>
> [3] (in-depth analysis on dataset shift’s impact) We thank the reviewer for the advice and have conducted new experiments to quantitatively analyze how dataset shifts have impact on the MIA performance. Basically, our experiments have two parts:
>
>
> - Quantifying dataset shifts (Section 5.1). We calculate three distance metrics between member and non-member datasets in our benchmark: normalized Wasserstein distance (NWD), Fréchet Distance (FD), and Mahalanobis Distance (MD). All distances are calculated based on CLIP-large. Among all setups covered, setup (a) and (b) in our benchmark suffer much bigger distances between members and non-members, for example, FDs of 0.32 and 0.24. This validates our conclusion that there are valid shifts between their members and non-members. Therefore, MIA methods could separate these two datasets according to the semantics rather than the membership, which raises the dataset shift concern in the paper. Setup (d) has medium distances between members and non-members, for example, a FD of 0.12. The distances are much smaller for Setup (c) and (e), for example, FDs < 0.10. This experiment quantitatively demonstrates the existence of dataset shifts and shows potential connections between dataset shifts and the MIA performance.
>
> - Relation between dataset shifts and MIA performance (Section 5.3). We construct a series of non-member datasets by mixing our two non-member datasets: COCO-val-2017 (with shifts to LAION) and LAION-MI (no shifts) with different proportions. We pick the proportions by 100% vs 0%, 75% vs 25%, 50% vs 50%, 25% vs 75%, and 0% vs 100%. We evaluate SecMI and PIA on these setups. The result in Figure 1 shows that there is positive correlation between the performance of these two MIA methods and the proportion of shifted non-members data. This shows that one can manipulate the result of MIA evaluation easily by only changing non-members, which is supposed to be irrelevant to the result and that current MIA evaluation is unreliable.
>
> If you have further questions, feel free to contact us.

---

> ### Author Response · Authors · 2024-12-02
> **Kindly request for further discussions**
>
> Dear Reviewer,
>
> We highly appreciate your constructive reviews that raised the concerns on the contribution of our paper. As we mentioned in the rebuttal, the main contribution of this paper is to alert a wrong trend of the current MIA research on diffusion models. While MIA benchmarks are not necessary to be defective, **all existing benchmarks** suffer from the two defects we presented and are continuously used in evaluating new methods. Hence, we believe it is necessary to point out the defects and provide a new fair benchmark. In addition, we conducted two experiments to quantitatively demonstrate how changing dataset shifts could manipulate the performance of diffusion MIAs. We believe these could effectively address your concerns in the advanced comprehension of dataset shifts. As the discussion phase is about to close, we will sincerely appreciate it if you can engage the discussion and provide further advice on our paper.

---

### Official Review · Reviewer_o9bC · 2024-11-04

**Soundness:** 3
**Presentation:** 3
**Contribution:** 3
**Rating:** 6
**Confidence:** 4

**Summary:**

The paper introduces a straightforward yet powerful benchmark to assess the performance of existing Membership Inference Attacks (MIA) on pre-trained diffusion models within the context of data authorization. The authors identified "overtraining" and "dataset shifts" as two significant limitations of current MIA methods. To address these issues, they developed a benchmark featuring five experimental setups.

**Strengths:**

- The writing is clear
- The structure is easy to follow
- The paper considered comprehensive comparison with the relate works

**Weaknesses:**

- I am unsure about the input for the membership inference attacks. In Lines 113-116, does x refer solely to the image, or is it a combination of the image and its prompt? I recommend that the authors clarify this in the problem setup.

- In Table 1, why does "LDM + CelebA" have $\times$ for both “Over-training” and “Shifted Datasets,” while in the bottom table, "LDM + CelebA" (i.e., the third row) has $\checkmark$ for both? Is this a typo, or have I misunderstood the notation?

- While I appreciate the authors’ efforts in benchmarking MIA methods in practical scenarios, I believe the paper’s analysis of the two challenges, “Over-training” and “Shifted Datasets,” could be more in-depth. For example, I recommend adding an analysis of how shifted datasets impact MIA performance based on the distance of non-members from the target data (e.g., considering extremely close, moderately distant, and far distant non-members).

**Questions:**

See Weakness part.

---

> ### Author Response · Authors · 2024-11-21
>
> We thank the insightful review and would like to address the issue by points:
>
>
> [1] (x refers to image or image+prompt) x solely refers to the image. We do not consider the prompt because 1) prompts could be highly variable and easily modified in training, thus not being reliable conditions for MIA and 2) None of baseline methods claim strong dependency on prompts. If future methods have this strong dependency, we would like to update our benchmark and take prompts into consideration. Also, we thank the reviewer for the advice and have fixed this in the problem definition.
>
>
> [2] (Typo in Table 1) Yes, it is a typo. We feel sorry for this typo and have fixed it in our new draft. "LDM + CelebA" has both dataset shift and over-training, for it uses two different datasets as members and non-members and trains the model for 500 epochs on the member dataset.
>
>
> [3] (in-depth analysis on dataset shift’s impact) We thank the reviewer for the advice and have conducted new experiments to quantitatively analyze how dataset shifts have impact on the MIA performance. Basically, our experiments have two parts:
>
> - Quantifying dataset shifts (Section 5.1). We calculate three distance metrics between member and non-member datasets in our benchmark: normalized Wasserstein distance (NWD), Fréchet Distance (FD), and Mahalanobis Distance (MD). All distances are calculated based on CLIP-large. Among all setups covered, setup (a) and (b) in our benchmark suffer much bigger distances between members and non-members, for example, FDs of 0.32 and 0.24. This validates our conclusion that there are valid shifts between their members and non-members. Therefore, MIA methods could separate these two datasets according to the semantics rather than the membership, which raises the dataset shift concern in the paper. Setup (d) has medium distances between members and non-members, for example, a FD of 0.12. The distances are much smaller for Setup (c) and (e), for example, FDs < 0.10. This experiment quantitatively demonstrates the existence of dataset shifts and shows potential connections between dataset shifts and the MIA performance.
>
> - Relation between dataset shifts and MIA performance (Section 5.3). We construct a series of non-member datasets by mixing our two non-member datasets: COCO-val-2017 (with shifts to LAION) and LAION-MI (no shifts) with different proportions. We pick the proportions by 100% vs 0%, 75% vs 25%, 50% vs 50%, 25% vs 75%, and 0% vs 100%. We evaluate SecMI and PIA on these setups. The result in Figure 1 shows that there is positive correlation between the performance of these two MIA methods and the proportion of shifted non-members data. This shows that one can manipulate the result of MIA evaluation easily by only changing non-members, which is supposed to be irrelevant to the result and that current MIA evaluation is unreliable.
>
>
>  Again, we thank the reviewer for the insightful review. If you have further questions, feel free to contact us.

---

> > ### Comment · Reviewer_o9bC · 2024-11-28
> > **Thank you for your reply**
> >
> > Thank you for your reply. The rebuttal effectively addresses my concerns, and I have adjusted my score to 6.

---

> > > ### Author Response · Authors · 2024-11-28
> > > **Thank you for raising the score**
> > >
> > > We are sincerely grateful for your providing constructive advice on our paper and considering our rebuttal. The updated analysis does complement our discussion on how dataset shifts impact the evaluation of diffusion MIA. Thank you!

---

### Author Response · Authors · 2024-11-21
**Updating a new draft of the paper**

We thank all reviewers for the insightful reviews and have accordingly updated a new draft of our paper. We list the updated points, noted as blue text in our new draft, as follows:

[1] Quantifying the dataset shift of benchmarks in CopyMark and showing the threshold between qualified / unqualified member and non-member datasets (Section 5.1).

[2] Relation between different levels of dataset shifts and MIA performances (Section 5.3).

[3] Fixing typos in Table 1.

[4] Brief guidelines on how to address the challenges identified on existing MIA methods under realistic scenarios (Appendix 3.3.1).

[5] Updating extra references.

[6] Explaining the difference between over-training in diffusion models and traditional over-fitting (Section 3).

---

### Author Response · Authors · 2024-11-30
**Kindly request for further discussions**

Dear Reviewers,

We highly appreciate your constructive review and appreciate your constructive advice. To address these concerns, we have updated our paper draft and conducted in-depth analysis on the impact of dataset shifts with extensive experiments, which further validates our assertion. As the rebuttal period draws to a close, we sincerely look forward to your further ideas on these concerns and kindly request your engagement in the discussion. We also want to hear your advice on the updated content. We would sincerely appreciate it if you could consider engaging in further discussions.

Best regards,

Authors of Submission7638

---

### Meta-Review · Area_Chair_DbKD · 2024-12-13

**Metareview:**

The paper aims to evaluate the state-of-the-art MIAs on diffusion models and reveal critical flaws and overly optimistic performance estimates in existing MIA evaluation. The presentation is clear and the experiments are also good. Reviewers like this paper, but find many issues that can not be easily fixed in a short time, such as the lack of discussion and the lack of the evaluation and so on.

**Additional Comments On Reviewer Discussion:**

Reviewers like this paper, but find many issues that can not be easily fixed in a short time, such as the lack of discussion and the lack of the evaluation and so on.

---

### Decision · Program_Chairs · 2025-01-22

Reject